# Let Images Give You More:
# Point Cloud Cross-Modal Training for Shape Analysis

**Xu Yan**[1,2†]**, Heshen Zhan**[1,2†]**, Chaoda Zheng**[1,2]**, Jiantao Gao**[4]**,**
**Ruimao Zhang**[3]**, Shuguang Cui**[2,1,5]**, Zhen Li**[2,1*]
[1]FNii, CUHK-Shenzhen, [2]SSE, CUHK-Shenzhen,
[3]SDS, CUHK-Shenzhen, [4]USV, Shanghai University, [5]Pengcheng Lab
{xuyan1@link.,heshenzhan@link.,lizhen@}cuhk.edu.cn

## Abstract

Although recent point cloud analysis achieves impressive progress, the paradigm of representation learning from a single modality gradually meets its bottleneck. In this work, we take a step towards more discriminative 3D point cloud representation by fully taking advantages of images which inherently contain richer appearance information, *e.g.,* texture, color, and shade. Specifically, this paper introduces a simple but effective point cloud cross-modality training (**PointCMT**) strategy, which utilizes view-images, *i.e.*, rendered or projected 2D images of the 3D object, to boost point cloud analysis. In practice, to effectively acquire auxiliary knowledge from view images, we develop a teacher-student framework and formulate the cross-modal learning as a knowledge distillation problem. PointCMT eliminates the distribution discrepancy between different modalities through novel feature and classifier enhancement criteria and avoids potential negative transfer effectively. Note that PointCMT effectively improves the point-only representation without architecture modification. Sufficient experiments verify significant gains on various datasets using appealing backbones, *i.e.*, equipped with PointCMT, PointNet++ and PointMLP achieve state-of-the-art performance on two benchmarks, *i.e.,* **94.4%** and **86.7%** accuracy on ModelNet40 and ScanObjectNN, respectively. Code will be made available at https://github.com/ZhanHeshen/PointCMT.

## 1 Introduction

As the fundamental 3D representation, point clouds have attracted increasing attention for various applications, *e.g.*, self-driving [2, 33, 34], robotics perception [7, 13, 6], *etc*. Generally, a point cloud consists of sparse and unordered points in the 3D space, which is significantly different from a 2D image with a dense and regular pixel array. Prior studies treat the understanding of 2D images and 3D point clouds as two separate problems, and both have their own merits and drawbacks. Concretely, rich color and fine-grained texture are easily obtained in 2D images, but they are ambiguous in depth and shape sensing. Previous works extract features on images through convolution neural networks (CNN). In contrast, point clouds are superior in providing spatial and geometric information but only preserve sparse and textureless features. Several prior studies process features on unstructured point clouds through local aggregation operators [34, 43]. It is natural to raise a question: *Could we use the rich information hidden in 2D images to boost 3D point cloud shape analysis*?

To address the above issue, one straightforward way is to leverage the benefits of both images and point clouds, *i.e.*, fusing information from two complementary representations with task-specific design [58, 24, 10, 32, 42]. However, utilizing additional image representation requires designing

---

*Corresponding author: Zhen Li. [†] Equal first authorship.

36th Conference on Neural Information Processing Systems (NeurIPS 2022).

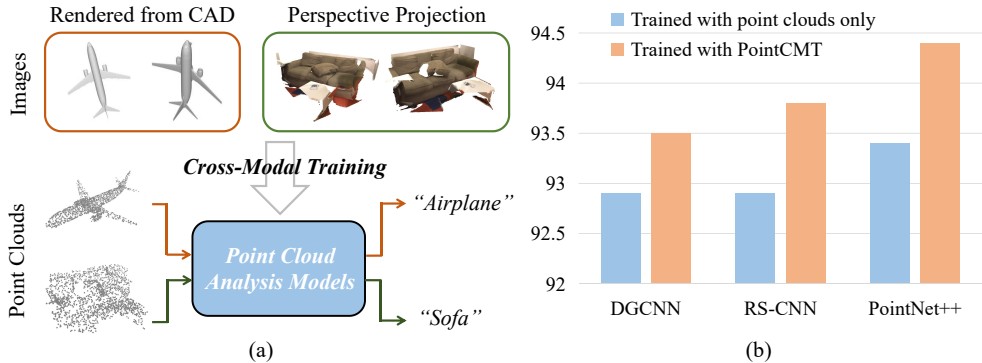

Figure 1: (a) Our proposed general Cross-Modal Training (PointCMT) strategy. It introduces priors from images into point cloud shape analysis models only in the training stage without any baseline model modification. (b) Classification accuracy (%) on ModelNet40 with or without training with our proposed PointCMT strategy. Noticeable improvements can be observed.

a multi-modal network, which takes the extra image inputs in both training and inference phases. Moreover, the exploiting extra-images is usually computation-intensive and paired-images are usually unavailable during inference. Thus, multi-modal learning meets its bottleneck in many aspects.

This paper tries to ease the barrier of cross-modal learning between images and point clouds. Inspired by knowledge distillation (KD) that achieves knowledge transfer from a teacher model to a student one, we formulate the cross-modal learning as a KD problem, conducting alignment between sample representations learned by images and point clouds. However, previous KD approaches usually assume that the training data used by the teacher and student are from the same distribution [17]. Thus, since sparse and disordered point clouds represent visual information different from images, naive feature alignment between two representations appeals to cause limited gains or negative transfer for the cross-modal scenario. To this end, we design a novel framework for cross-modal KD and propose the point cloud cross-modal training strategy, *i.e.*, **PointCMT** in Figure 1 (a), which distills features derived from images into the point cloud representation. Specifically, multiple view images for each 3D object can be generated through either rendering the CAD model or conducting perspective projection on the point cloud from different viewpoints. These free auxiliary images are fed into the image network to obtain the global representations for the object. Besides, feature and classifier enhancements are conducted between the point cloud and image features, in which the newly proposed criteria effectively avoid negative transfer between different modalities, *i.e.*, directly applying [17] hampers the performance on ModelNet40. After training, the model gains higher performance, only taking point clouds as input without architecture modification.

Compared with multi-modal approaches, our solution has the following preferable properties: **1) Generality**: It can be integrated with arbitrary point cloud analysis models without structural modification. **2) Effectively**: It significantly boosts the performance upon several baseline approaches, *e.g.*, Point-Net++ [34] achieves state-of-the-art 94.4% from 93.4% overall accuracy on ModelNet40, as shown in Figure 1 (b). **3) Efficiency**: Our PointCMT only utilizes auxiliary image data in the training stage. After training, the enhanced 3D model infers without image inputs. **4) Flexibility**: The extensive experiments illustrate that PointCMT performs superior even without colorized and dedicated rendered images, *i.e.*, it can also greatly improve the performance when using images directly projected by sparse and textureless point clouds. Thus, it provides an alternative solution to enhance the point cloud shape analysis when the additional rendered images are not accessible.

In summary, our contributions are: **1)** This paper formulates cross-modal learning on point cloud analysis as a knowledge distillation problem, where we utilize the merits of texture and color-aware 2D images to acquire more discriminative point cloud representation. **2)** We propose point cloud cross-modal training, *i.e.,* PointCMT, strategy with corresponding criteria to boost point cloud models during the training stage. **3)** Extensive experiments on several datasets verify the effectiveness of our approach, where PointCMT greatly boosts several baseline models even on the state-of-the-art, *e.g.*, PointNet++ [34] trained with PointCMT gains **1.0%** and **4.4%** accuracy improvements on ModelNet40 [48] and ScanObjectNN, respectively. Even based upon PointMLP [31], it increases its accuracy by 1% to **86.7%** on ScanObjectNN dataset.

## 2 Related Works

**3D Shape Recognition Based on Point Clouds.** These stream methods directly process raw point clouds as input (also called *point-based methods*). They are pioneered by PointNet [33], which approximates a permutation-invariant set function using a per-point Multi-Layer Perceptron (MLP) followed by a max-pooling layer. Later on, point-based methods aim at designing local aggregation operators for local feature extraction. Specifically, they generally sample multiple sub-points from the original point cloud, and then aggregate neighboring features of each sub-point through local aggregation operators, in which point-wise MLPs [34, 31], adaptive weight [44, 46, 27] and pseudo grid based methods [39, 20] are proposed. More recently, there are some attempts to utilize non-local operator [54], or Transformer [60, 14] to mine the long distance dependency. This paper also follows the paradigm of point-based methods to conduct point cloud shape analysis.

**3D Shape Recognition Based on Images.** Since point clouds are irregular and unordered, some works consider projecting the 3D shapes into multiple images from different viewpoints (also called *view-based methods*) and then leverage the well-developed 2D CNNs to process 3D data. One seminal work of multi-view learning is MVCNN [38]. It extracts per-view features with a shared CNN in parallel, then aggregates via a view-level max-pooling layer. Most follow-up works propose more effective modules to aggregate the view-level features. For instance, some of them enhance the aggregated feature by considering similarity among views [11, 56] while others focus on the viewpoint relation [45, 22]. The above methods usually utilize ad-hoc rendered images for each 3D shape, including shade and texture for the surface mesh. Therefore, they generally achieve higher performance than point-based methods using sparse point clouds as input. Recently, [12] proposes a simple but effective method (*SimpleView*) through directly projecting sparse point clouds onto image planes, achieving comparable performance with point-based methods. Inspired by view-based methods, this paper takes advantage of extracted image features from the view-based method, which are utilized as prior knowledge to boost point cloud shape analysis.

**Knowledge Distillation.** Knowledge distillation (KD) aims at compressing a large network (teacher) to a compact and tiny one (student), and boost the performance of the student at the same time. The concept was first shown by Hinton *et al.* [17], which trains a student by using the softened logits of a teacher as targets. Over the past few years, several subsequent approaches [23, 1, 5, 40, 9, 61] use different criteria to align the sample representations between the teacher and student. However, almost all the existing works assume that the training data used by the teacher and student networks are from the same distribution. Our experiment illustrates that new biases and negative transfer will be introduced in the distillation process if cross-modal data from different distribution (*e.g.*, features extracted from unordered point cloud and regular grid image) is utilized directly on previous KDs.

**Cross-Modal Knowledge Transfer.** Cross-modal knowledge transfer in computer vision is a relatively emerging field that aims to utilize additional modalities at the training stage and enhance the model's performance on the target modality at the inference. Recently, there are 3D-to-2D knowledge transfer approaches, which adopt geometric aware 3D features from point clouds to enhance the performance of 2D tasks through a contrastive manner [18] or feature alignment [30]. Later on, approaches attempt to transfer priors in images to enhance 3D point cloud-related tasks, and some are designed for specific tasks. Concretely, [28, 26] propose the images-to-point contrastive pre-training, [50] inflats 2D convolution kernels to the 3D ones and [57, 59, 53] independently apply cross-modal training for visual grounding, captioning and semantic segmentation. Inspired but different from the above, we are the first to conduct image-to-point knowledge distillation for point cloud analysis.

## 3 Methodlogy

### 3.1 Problem Statement

Let $\mathcal{P} \in \mathbb{R}^{N \times 3}$ and $y \in \mathbb{R}^1$ be the point cloud and ground-truth label of the 3D object. Its corresponding view-image counterparts can be denoted as $\mathcal{I} \in \mathbb{R}^{V \times H \times W \times 3}$, where $N$, $V$ and $(H, W)$ are the number of points, number of view-images and image size, respectively. View images can be gained by rendering the 3D CAD model [38] or perspective projecting the raw point cloud [12]. We denote $T$ and $S$ as the image and point cloud analysis networks, respectively, and regard them as the **teacher** and **student** in traditional knowledge distillation (KD). For these networks, we split each of them into two parts: (i) **Encoders** (*i.e.*, feature extractors $\text{Enc}^{img}(\cdot)$ and $\text{Enc}^{pts}(\cdot)$), the output of

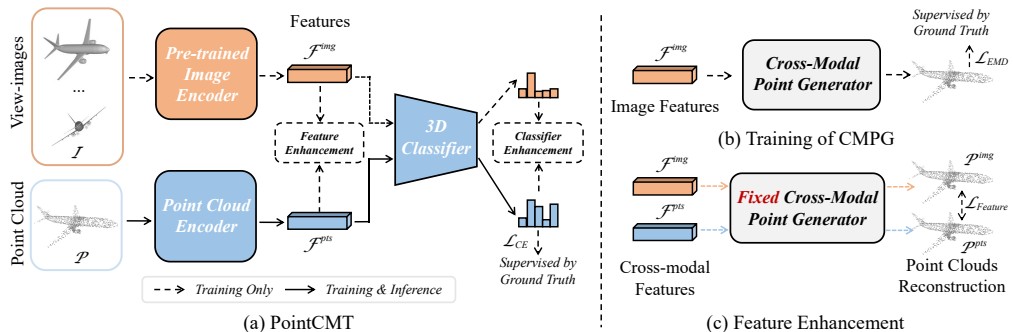

Figure 2: **(a)** The architecture of PointCMT. Multiple view images are gained through rendering the 3D CAD model or perspective projecting the raw point cloud, and the pre-trained image network distills the knowledge to the point cloud analysis network via two matching processes. The first is feature enhancement, which aligns features through a pre-trained cross-modal point generator through the process in (b). The second one is classifier enhancement, which aligns the output distribution of the point classifier by taking cross-modal features as inputs. **(b)** Training process of cross-modal point generator (CMPG). **(c)** Illustration of feature enhancement.

which at the last layer are global feature representations $\mathcal{F}^{img} \in \mathbb{R}^D$ and $\mathcal{F}^{pts} \in \mathbb{R}^D$. (ii) **Classifiers**, which project the feature representation $\mathcal{F}^{img}$ and $\mathcal{F}^{pts}$ into class logits through $\mathrm{Cls}^{img}(\mathcal{F}^{img})$ and $\mathrm{Cls}^{pts}(\mathcal{F}^{pts})$, where $\mathrm{Cls}(\cdot)$ denotes the classifier.

Since we formulate the cross-modal learning as a KD problem, its goal in the learning process is to distill the priors knowledge from the image into point cloud features, obtaining an image-enhanced ideal feature $\mathcal{F}^{KD}$. During the knowledge distillation, we parameterize the teacher and student networks with $\theta_T$ and $\theta_S$, and denote the knowledge of the teacher network as $\mathcal{K}_T$. From the Bayesian perspective, a neural network can be viewed as a probability model, *e.g.*, $P(y|\mathcal{P}, \theta_S)$ for point cloud analysis model as an example: given an input point cloud $\mathcal{P}$, the network assigns an output probability with the parameters $\theta_S$. Therefore, if we want the student network guided by image knowledge on the input sample, our goal can be further reformulated as maximizing the probability $\mathcal{P}(\mathcal{F}^{KD}|\mathcal{P}, y; \theta_S, \mathcal{K}_T)$, which can also be used to measure the ability of student network extracting the feature with cross-modal information. To find the lower bound of the above probability, we define the discrepancy $g$ between theoretically discriminative features $\mathcal{F}^{img}_*$ and $\mathcal{F}^{pts}_*$ as

$$g = \mathcal{P}(\mathcal{F}^{img}_*|\mathcal{I}, y; \theta_T) - \mathcal{P}(\mathcal{F}^{pts}_*|\mathcal{P}, y; \theta_S, \mathcal{K}_T), \tag{1}$$

where $\mathcal{F}^{img}_*$ and $\mathcal{F}^{pts}_*$ are ideal features in specific modalities. In Lemma 1, we give the lower bound of cross-modal learning as a KD problem. The proof is provided in the supplementary material.

**Lemma 1**: *By the definition above, $\mathcal{P}(\mathcal{F}^{KD}|\mathcal{P}, y; \theta_S, \mathcal{K}_T)$ is bounded below by $\mathcal{P}(\mathcal{F}^{KD}|\mathcal{I}, y; \theta_T) + \lambda - g$, where $\lambda$ is*

$$\lambda = \mathcal{P}(\mathcal{F}^{img}_*|\mathcal{I}, y; \theta_T) - \mathcal{P}(\mathcal{F}^{KD}|\mathcal{I}, y; \theta_T). \tag{2}$$

In Lemma 1, $\lambda$ measures the compatibility of knowledge distillation of the image networks, and can be viewed as a constant when the architectures are determined. $\mathcal{P}(\mathcal{F}^{KD}|\mathcal{I}, y; \theta_T)$ is also a constant when the parameters $\theta_T$ and the architecture of the point cloud analysis model are fixed, *e.g.,* using a pre-trained image network. Therefore, the Lemma ensures that during the knowledge distillation, one can maximize $\mathcal{P}(\mathcal{F}^{KD}|\mathcal{P}, y; \theta_S, \mathcal{K}_T)$ through minimizing $g$, which gives a theoretical guarantee for the KD problem.

If we adopt previous KD studies in cross-modal scenario, they minimize $g$ through making student directly approximate the teacher's features [23, 1, 5] or predict logits [17]. However, in a common KD problem, the teacher and student are generally trained on the same dataset with an identical distribution [17]. Moreover, the teacher generally achieves better performance than the student. In contrast, the image and point cloud analysis models tend to learn different feature representations and logits distribution which are generally complementary. Direct alignment may cause a limited gain or even negative transfer. Moreover, previous KD approaches treat encoders and classifiers as a whole architecture since the teacher and student networks generally have the same components. However, the point cloud convolution in encoder is significantly different from the 2D CNN, but they have the same classifier design.

---

**Algorithm 1:** Process in point cloud cross-modal training (PointCMT)

---

**Data:** The point clouds $\{\mathcal{P}_i\}_{i=1}^M$, corresponding ground-truth labels $\{y_i\}_{i=1}^M$ and view-images set $\{\mathcal{I}_i\}_{i=1}^M$, where $M$ is a number of training data samples.

**Stage I: Training image encoder and image classifier:** Taking view-images set $\{\mathcal{I}_i\}_{i=1}^M$ as input, image encoder produce the image features $\{\mathcal{F}_i^{img}\}_{i=1}^M$, which are then fed into image classifier and obtain prediction logits $\{\mathrm{Cls}^{img}(\mathcal{F}^{img})\}_{i=1}^M$ supervised by ground-truth labels $\{y_i\}_{i=1}^M$.

**Stage II: Training cross-modal point generator:** As shown in Figure 2 (b), the generator takes features $\{\mathcal{F}_i^{img}\}_{i=1}^M$ as input, reconstructs point cloud $\{\hat{\mathcal{P}}_i^{img}\}_{i=1}^M$ and supervised by point clouds $\{\mathcal{P}_i\}_{i=1}^M$.

**Stage III: Image priors assisted training:** The point encoder takes point clouds $\{\mathcal{P}_i\}_{i=1}^M$ as input and generate point features $\{\mathcal{F}_i^{pts}\}_{i=1}^M$. The point classifier gains prediction logits $\{\mathrm{Cls}^{pts}(\mathcal{F}^{pts})\}_{i=1}^M$ through taking features $\{\mathcal{F}_i^{pts}\}_{i=1}^M$. The feature enhancement align cross-modal features $\{\mathcal{F}_i^{img}\}_{i=1}^M$ and $\{\mathcal{F}_i^{pts}\}_{i=1}^M$ through **Feature Enhancement**, and **Classifier Enhancement** enhance the point classifier via matching $\{\mathrm{Cls}^{pts}(\mathcal{F}^{pts})\}_{i=1}^M$ and $\{\mathrm{Cls}^{pts}(\mathcal{F}^{img})\}_{i=1}^M$.

---

To solve the cross-modal KD problem, we propose PointCMT, which effectively solves the cross-modal learning problem. The workflow of PointCMT is demonstrated in Figure 2 (a) and can be summarized as Algorithm 1. Specifically, there are three stages exist in PointCMT. In Stage I (Section 3.2), we train the image encoder and classifier using view-images and ground-truth labels. In Stage II (Section 3.3), we train the cross-modal point generator (CMPG) through image features, and we independently align features and logits of two modalities in stage III (Section 3.4), where the encoder and classifier of point cloud analysis network are both enhanced.

## 3.2 Learning Image Priors

For each 3D object, we use multiple view-images (*i.e.*, rendered color images or projected images from raw point cloud) as additional data, and the whole process can be described as:

$$\mathcal{F}^{img} = \mathcal{A}\{\mathrm{CNN}(\mathcal{I}_v)\}_{v=1}^V. \tag{3}$$

Inspired by view-based learning approaches [38], $V$ view-images from $\mathcal{I}$ flow into a shared-weights image feature extractor $\mathrm{CNN}(\cdot)$ to obtain a series of vectors. By aggregating the view-level vectors via an aggregation function $\mathcal{A}\{\cdot\}$, we obtain a global feature representation $\mathcal{F}^{img}$ from all images, which integrates shape information from multiple views. Finally, an image classifier maps the above global feature to gain a prediction logits through $\mathrm{Cls}^{img}(\mathcal{F}^{img})$, which is supervised by the ground truth label $y$ through cross-entropy loss $\mathcal{L}_{CE}$.

## 3.3 Cross-Modal Point Generator

A cross-modal point generator (CMPG) can be seen as a nonlinear transformation $\mathbb{R}^D \to \mathbb{R}^{N \times 3}$ that maps the global feature representation $\mathcal{F}^{img}$ acquired from images into the Euclidean space. Thus, it can avoid potential negative transfer effectively by directly aligning cross-modal features from different distributions. In order to better learn image priors in the point cloud analysis network, we pre-train the CMPG through the image feature $\mathcal{F}^{img} \in \mathbb{R}^D$, and fix it in the distillation stage. Figure 2 (b) illustrates the pre-training stage of CMPG. It takes the image feature as input and reconstruct a point cloud $\hat{\mathcal{P}}^{img} \in \mathbb{R}^{N \times 3}$, which is supervised by the original point cloud $\mathcal{P}$ through Earth Mover's distance (EMD) [37]:

$$\mathcal{L}_{\mathrm{EMD}}(\mathcal{P}, \hat{\mathcal{P}}^{img}) = \min_\phi \sum_{p \in \mathcal{P}} ||p - \phi(p)||, \tag{4}$$

where $|\mathcal{P}| = |\mathcal{P}^{img}|$ and $\phi : \mathcal{P} \to \hat{\mathcal{P}}^{img}$ is a bijection, *i.e.*, for each point $p \in \mathcal{P}$, $\phi(\cdot)$ finds a sole point correspondence in $\hat{\mathcal{P}}^{img}$. After pre-training, CMPG reconstructs a point cloud through an image-relative feature representation.

## 3.4 Image Priors Assisted Training

During the training stage, three objectiveness should be optimized:

**Classification Loss.** In our PointCMT, arbitrary point cloud analysis models can be assembled. Generally, it should be designed through a point encoder and a classifier, where the point encoder takes a point cloud as input, generates the point cloud feature representation $\mathcal{F}^{pts}$ and feeds it into the classifier $\text{Cls}^{pts}(\cdot)$ to obtain a class logits. Finally, the cross-entropy loss $\mathcal{L}_{CE}$ is used as criteria.

**Feature Enhancement Loss.** Unlike previous KD methods that directly align features of teacher and student, we first transform the cross-modal features into Euclidean space. As shown in Figure 2 (c), the pre-trained CMPG independently transform $\mathcal{F}^{img}$ and $\mathcal{F}^{pts}$, obtaining two point clouds $\hat{\mathcal{P}}^{img}$ and $\hat{\mathcal{P}}^{pts}$, respectively. After that, EMD loss is conducted on the two-point clouds as an objectiveness:

$$\mathcal{L}_{\text{Feature}} = \mathcal{L}_{\text{EMD}}(\hat{\mathcal{P}}^{pts}, \hat{\mathcal{P}}^{img}) = \min_{\phi} \sum_{p \in \hat{\mathcal{P}}^{pts}} ||p - \phi(p)||, \tag{5}$$

where $|\hat{\mathcal{P}}^{pts}| = |\mathcal{P}^{img}|$ and $\phi : \hat{\mathcal{P}}^{pts} \to \hat{\mathcal{P}}^{img}$ is a bijection. Compared with traditional $L_2$ loss, the EMD distance is natural for solving an assignment problem for permutation-invariant point sets. For all but a zero-measure subset of point set pairs, the optimal bijection $\phi$ is unique and invariant under the infinitesimal movement of the points. Thus, EMD is differentiable almost everywhere.

**Classifier Enhancement Loss.** In addition to supervising the point feature extractor through the above Feature Enhancement, we further propose constraints conducted on the point classifier (as Figure 2 (a)). Specifically, the image feature generated by the teacher network is fed into the point classifier, in which the gradient only back-propagates to the point classifier. Besides, classifier enhancement is proposed to enable the point classifier to handle the image feature during the distillation. It aligns outputs logits of the point classifier by using image and point features as inputs. This constraint is modified based on the distillation loss in Hinton *et al.* [17] as in Equation (6), but in this case, two sets of logits come from the same classifier.

$$\mathcal{L}_{\text{Hinton}} = \mathcal{D}_{KL}(\text{Cls}^{img}(\mathcal{F}^{img}) || \text{Cls}^{pts}(\mathcal{F}^{pts})), \tag{6}$$

where $\mathcal{D}_{KL}(\cdot || \cdot)$ is KL divergence. In contrast, our proposed classifier enhancement is more suitable for the cross-modal scenario where a great discrepancy exist between image and point cloud features. Concretely, the loss for classifier enhancement can be written as

$$\mathcal{L}_{\text{Classifier}} = \mathcal{D}_{KL}(\text{Cls}^{pts}(\mathcal{F}^{img}) || \text{Cls}^{pts}(\mathcal{F}^{pts})). \tag{7}$$

**Final Loss.** The final loss is a weighted sum of the above three losses $\mathcal{L} = \mathcal{L}_{CE} + \alpha \mathcal{L}_{\text{Feature}} + \beta \mathcal{L}_{\text{Classifier}}$, where $\alpha = 30$ and $\beta = 0.3$ are the weights to adjust the ratios of each loss, respectively.

## 4 Experiment

### 4.1 Shape Classification on ModelNet40

We firstly evaluate our PointCMT on synthetic dataset ModelNet40 [48], which is a large-scale 3D CAD model dataset.

**Dataset Description and Processing.** ModelNet40 is composed of 9,843 train models and 2,468 test models in 40 classes. For the input of the 3D network, we use point clouds provided by the official dataset, which is the same as PointNet [33]. For the input of the image network, we use 20 rendered view images from CAD models utilized in RotationNet [22]. These images have a resolution of $224 \times 224$. Since they consider both the mesh surfaces and illumination, they can provide more information to the 3D network. Selected samples of point clouds and corresponding multi-view images are shown in Figure 3.

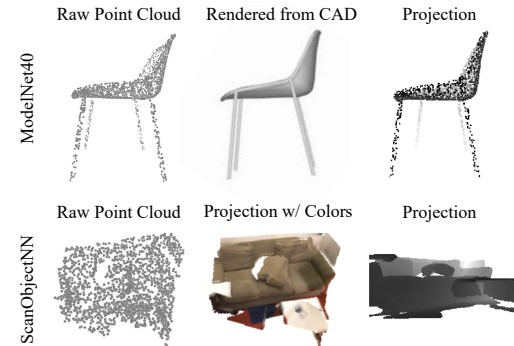

Figure 3: Different view-image generation strategies used in our experiment.

**Implementation.** For image network, we use ResNet-18 [16] pre-trained on ImageNet [8] as the feature extractor. Following MVCNN [38], we obtain the global shape feature by applying view-wise max-pooling to the view-level features. Finally, a fully-connected layer is used to output the

Table 1: Classification results on ModelNet40 dataset. With only 1k points, PointNet++ trained with PointCMT achieves state-of-the-art results on both class mean accuracy (mAcc) and overall accuracy (OA) metrics. Here, 'pnt' and 'nor' denote points and normal vectors, respectively. The speed (samples/second) tested on one Tesla V100 GPU and four cores AMD EPYC 7351@2.60GHz CPUs, where † denotes the results from the original paper. * For PointNet++, we train it with protocol of RS-CNN [27] as mentioned in [12]. The best and second best are marked in **bold** and underline.

| Method | Input | #Points | mAcc(%) | OA(%) | Speed | Param. |
|---|---|---|---|---|---|---|
| PointNet [33] | pnt | 1k | 86.0 | 89.2 | - | 3.47M |
| PointNet++ [34] | pnt, nor | 5k | - | 91.9 | - | 1.47M |
| PointCNN [25] | pnt | 1k | 88.0 | 92.5 | - | - |
| PointConv [47] | pnt, nor | 1k | - | 92.5 | $80^\dagger$ | 18.6M |
| KPConv [39] | pnt | 7k | - | 92.9 | $10^\dagger$ | 15.2M |
| PointASNL [54] | pnt, nor | 1k | - | 93.2 | - | - |
| PosPool [29] | pnt | 5k | - | 93.2 | - | - |
| Point Transformer [60] | pnt | 1k | 90.6 | 93.7 | - | - |
| GBNet [36] | pnt | 1k | 91.0 | 93.8 | $112^\dagger$ | 8.4M |
| GDANet [51] | pnt | 1k | - | 93.8 | $14^\dagger$ | 0.9M |
| SimpleView [12] | pnt | 1k | - | 93.9 | 2208 | 1.64M |
| CurveNet [49] | pnt | 1k | - | 94.2 | $15^\dagger$ | 2.0M |
| PointMLP [31] | pnt | 1k | **91.4** | **94.5** | 139 | 12.6M |
| DGCNN [43] (baseline) | pnt | 1k | 90.2 | 92.9 | 518 | 1.68M |
| RS-CNN [27] (baseline) | pnt | 1k | 89.3 | 92.9 | 2174 | 1.17M |
| PointNet++ [34] (baseline) | pnt | 1k | 90.1 | 93.4* | 300 | 1.62M |
| DGCNN **w/** PointCMT | pnt | 1k | 90.8 (+0.6) | 93.5 (+0.6) | 518 | 1.68M |
| RS-CNN **w/** PointCMT | pnt | 1k | 90.1 (+0.8) | 93.8 (+0.9) | 2174 | 1.17M |
| PointNet++ **w/** PointCMT | pnt | 1k | 91.2 (+1.1) | 94.4 (+1.0) | 300 | 1.62M |

classification logits. During the training process of the above network, we use SGD as our optimizer with a learning rate of 0.01. The batch size is set to 128 for 50 epochs. After that, we fix the image network and train the CMPG with Adam and 32 batch size for 50 epochs. In practice, CMPG consists of three-layer MLP. For the point cloud analysis models, DGCNN [43] and RS-CNN [27] are independently trained with the training strategies provided in their official codes. PointNet++ [34] is trained with strategy of RS-CNN [27] as [12] for a better performance.

**Comparison with State-of-the-arts.** The classification results on ModelNet40 are shown in Table 1, where the overall accuracy (OA) and class mean accuracy (mAcc) are compared. The upper part of the table illustrates the results of current state-of-the-art methods, in which we use PointNet++ [34], RS-CNN [27] and DGCNN [43] as our baselines. We do not use PointMLP [31] as our baseline since it cannot robustly reproduce the highest results on ModelNet40, where the issue is mentioned in their open-sourced codes. For models trained from scratch, PointMLP [31] achieves the highest accuracy. As shown in the lower part of the table, after training with PointCMT, the performance of all baselines is greatly boosted, *i.e.,* 1.0% improvement upon PointNet++ and 0.9% for RS-CNN and 0.6% for DGCNN. We also compare our method to several open-sourced methods and report the parameters and testing speed. As shown in the last two columns of the table, though PointMLP gains 0.1 higher overall accuracy, its network consists of about 7.7× parameters and only achieves 46% speed of PointNet++. In contrast, PointCMT performs well on light-weighted models, which shows its great potential for real-time applications, *e.g.*, scene parsing in autonomous driving.

## 4.2 Shape Classification on ScanObjectNN

Though ModelNet40 is the widely used benchmark for point cloud analysis, it may not meet the realistic requirement due to its synthetic nature. To this end, we also conduct experiments on the ScanObjectNN benchmark [41], which is a real-world dataset.

**Dataset Description and Processing.** ScanObjectNN collects 2,902 objects from real-world indoor scenes ScanNet [7] and SceneNN [19], categorizing into 15 categories. Several variants are provided in the dataset, where the most challenging one is PB_T50_RS, *i.e.*, introducing perturbation objects (11,416 and 2,882 data for training and test) via random translation, shift, rotation and scaling. Due to background, noise, and occlusions, this benchmark poses significant challenges to existing point cloud

Table 2: Classification on ScanObjectNN. We examine all methods on original objects (OBJ_ONLY) and more challenging variant (PB_T50_RS). We train PointNet++ and PointMLP with protocol of [12] for a fair comparison. The best and second best are marked in **bold** and underline. We train and test for four runs and report mean ± std results.

| Method | OBJ_ONLY | | PB_T50_RS | |
|---|---|---|---|---|
| | mAcc(%) | OA(%) | mAcc(%) | OA(%) |
| 3DmFV [3] | - | 73.8 | 58.1 | 63.0 |
| PointNet [33] | - | 79.2 | 63.4 | 68.2 |
| SpiderCNN [52] | - | 79.5 | 69.8 | 73.7 |
| PointNet++ [34] | - | 84.3 | 75.4 | 77.9 |
| DGCNN [43] | - | 86.2 | 73.6 | 78.1 |
| PointCNN [25] | - | 85.5 | 75.1 | 78.5 |
| DRNet [35] | - | - | 78.0 | 80.3 |
| GBNet [36] | - | - | 77.8 | 80.5 |
| SimpleView [12] | 86.2 | 89.0 | - | 80.8 |
| PRANet [4] | - | - | 79.1 | 82.1 |
| MVTN [15] | - | - | - | 82.8 |
| PointNet++ [34] (baseline) | 85.4±0.2 | 87.4±0.1 | 75.5±0.3 | 79.2±0.2 |
| PointMLP [31] (baseline) | 89.1±0.3 | 92.2±0.3 | 83.9±0.5 | 85.4±0.3 |
| PointNet **w/** PointCMT | 89.0±0.3 (+3.7) | 91.6±0.2 (+4.3) | 79.9±0.3 (+4.4) | 83.1±0.2 (+3.9) |
| PointMLP **w/** PointCMT | **91.8±0.2** (+2.6) | **93.2±0.3** (+1.0) | **84.4±0.4** (+0.4) | **86.4±0.3** (+1.0) |

analysis methods. Furthermore, since the PB_T50_RS dataset only preserves the spatial coordinates (XYZ) for each object while the other information, such as RGB, is discarded, we also compared the original 2,902 objects (OBJ_ONLY), which includes additional RGB information. On both above datasets, we only use depth images through conducting perspective projection on raw point cloud as additional inputs, as shown in the last column in Figure 3. Section 4.6 will discuss more results using projection with additional color information.

**Implementation.** All view-images in ScanObjectNN are gained by the projection of raw point clouds, and we follow the structure of [12] only generate six images for PB_T50_RS and OBJ_ONLY. We train the image network from scratch with batch size 32 and SGD optimizer for longer epochs of 1,000. The training strategy of CMPG is the same as ModelNet40. For both point cloud models trained from scratch and with PointCMT, we use SGD optimizer for 1,000 epochs with batch size 32.

**Comparison with State-of-the-arts.** The results are shown in Table 2, where PointNet++ [34] and current state-of-the-art PointMLP [31] are chosen as our baselines. PointCMT significantly improves the performance on both class mean accuracy (mAcc) and the overall accuracy (OA), even on state-of-the-art methods. Specifically, although background, noise, and occlusions exist on PB_T50_RS dataset, PointCMT still improves the overall accuracy of PointNet++ by 3.9%. Moreover, PointCMT also achieves state-of-the-art results on OBJ_ONLY dataset. Note that there is no auxiliary information provided in images, and all view images are generated through the perspective projection of points coordinates. Nevertheless, PointCMT still dramatically increases the mAcc of PointMLP from 89.4% to 92.0% (+2.6%).

## 4.3 Data Efficient Learning

We evaluate our approach under limited data scenarios in Table 3. Here, we only sample a small amount of training data in each category on ModelNet40, and evaluate the entire testing data. Our PointCMT shows an even more significant gap when using a small subset of the training data, again compared to Point-Net++ which trained from scratch. Es-

Table 3: Data efficient learning on ModelNet40. We train PointNet++ [34] with a small amount of training data and train with PointCMT.

| Data percentage | Train from scratch | w/ PointCMT |
|---|---|---|
| 2% | 73.3 | **75.2** (+1.9) |
| 5% | 82.1 | **83.5** (+1.4) |
| 10% | 85.1 | **87.9** (+2.8) |
| 20% | 88.4 | **89.3** (+0.9) |

pecially when facing only 2% and 10% of the training data, we achieve about 1.9 and 2.8% improvements, respectively. This result illustrates that PointCMT provides more vital guidance for point cloud models in the low data regime.

Table 4: Ablation study on ModelNet40 and ScanObjetNN datasets. Overall accuracy (%) as metrics.

| Model | FE | CE | ModelNet40 | OBJ_ONLY | PB_T50_RS |
|-------|----|----|-----------|----------|-----------|
| PointNet++ | ✗ | ✗ | 93.4 | 87.5 | 79.4 |
| | ✓ | ✗ | 93.8 (+0.4) | 89.2 (+1.7) | 82.5 (+3.1) |
| | ✗ | ✓ | 94.0 (+0.6) | 91.3 (+3.8) | 82.3 (+2.9) |
| | ✓ | ✓ | 94.4 (+1.0) | 91.8 (+4.3) | 83.3 (+3.9) |

## 4.4 Ablation study

The ablation results on three datasets are summarized in Table 4, in which we use PointNet++ as our baselines. We first test the effectiveness of feature enhancement (FE) and classifier enhancement (CE) in PointCMT. The results demonstrate that only using FE already significantly boosts the performance on both datasets, *i.e.*, increases the overall accuracy by 0.4% and 3.1% on ModelNet40 and ScanObjetNN. Only using classifier enhancement (CE) improves the accuracy by around 0.6% and 2.9%. Finally, it is surprising that when we use both FE and CE during the training phase, it achieves the best result of 94.4% and 83.3%, respectively.

## 4.5 Comparison with Knowledge Distillation Methods

To further verify the effectiveness of our proposed method compared with typical teach-student architecture and other knowledge distillation manners, we compare PointCMT with typical approaches of knowledge transfer in Table 5. Among all the methods, Hinton *et al.* [17] is a pioneer study for knowledge distillation, while Huang *et al.* [21] and Yang *et al.* [55] are recent works. As shown in the table, directly aligning features

Table 5: Comparison with knowledge distillation methods. We compare overall accuracy (OA,%) gained by PointNet++ on ModelNet40 and ScanObjectNN.

| Method | ModelNet40 | PB_T50_RS |
|--------|-----------|-----------|
| Baseline | 93.4 | 79.4 |
| Hinton *et al.* [17] | 93.1 (-0.3) | 81.8 (+2.4) |
| Huang *et al.* [21] | 93.6 (+0.2) | 82.0 (+2.6) |
| Yang *et al.* [55] | 93.9 (+0.5) | 81.1 (+1.7) |
| PointCMT (ours) | **94.4** (+1.0) | **83.3** (+3.9) |

as [17] between two modalities will cause a negative transfer on ModelNet40. This phenomenon does not appear on ScanObjectNN, since view images projected via point clouds may have a smaller gap than rendered images of CAD models. Nevertheless, other KD techniques only achieve marginal improvement compared with PointCMT.

## 4.6 Different View-image Generation

In this section, we compare results with different view-image generation strategies. As shown in Figure 3, multiple view-image types can be applied in our framework, and we compare the results in Table 6. As illustrated in the table, images rendered from the CAD model

Table 6: Results through different view-image generation. We compare overall accuracy (OA,%) on ModelNet40 and ScanObjectNN OBJ_ONLY datasets.

| View-images | ModelNet40 | OBJ_ONLY |
|-------------|-----------|----------|
| Rendered from CAD | **94.4** | - |
| Projection | 94.0 | **91.8** |
| Projection w/ color | - | 90.7 |

improve more compared with only using projection since the former provides additional shade and texture information. In contrast, we find out that using additional colors in the OBJ_ONLY dataset cannot boost performance. The reason is that OBJ_ONLY dataset only contains 2,902 objects, and image networks are easier to overfit when using the color information.

## 5 Conclusion

In this work, we propose a point cloud cross-modal training strategy named PointCMT. By exploiting some sophisticated architecture and reasonable criteria function, our PointCMT can boost the performance significantly for point cloud analysis methods on several benchmarks, outperforming previous methods by a large margin. We believe that our work can be applied to a broader range of other scenarios in the future, such as 3D semantic segmentation and object detection. Meanwhile, our method provides an alternative solution to the comprehension of 3D scenes with severe texture details missing. It can improve performance through image priors and knowledge transfer.

## Acknowledgment

This work was supported in part by NSFC-Youth 61902335, by the Basic Research Project No. HZQB-KCZYZ-2021067 of Hetao Shenzhen HK S&T Cooperation Zone, by the National Key R&D Program of China with grant No.2018YFB1800800, by Shenzhen Outstanding Talents Training Fund, by Guangdong Research Project No. 2017ZT07X152 and No. 2019CX01X104, by the Guangdong Provincial Key Laboratory of Future Networks of Intelligence (Grant No. 2022B1212010001), by the NSFC 61931024&8192 2046, by NSFC-Youth 62106154, by zelixir biotechnology company Fund, by Tencent Open Fund, and by ITSO at CUHKSZ.

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
