# Supplementary Material

## A  Theoretical Proof

We provide the detailed theoretical proof for the lemmas proposed in the main manuscript. Define the discrepancy $g$ between the discriminative image and point cloud features as

$$g = \mathcal{P}(\mathcal{F}_*^{img}|\mathcal{I}, y; \theta_T) - \mathcal{P}(\mathcal{F}_*^{pts}|\mathcal{P}, y; \theta_S, \mathcal{K}_T), \tag{1}$$

**Lemma 1**: *By the definition above, $\mathcal{P}(\mathcal{F}^{KD}|\mathcal{P}, y; \theta_S, \mathcal{K}_T)$ is bounded below by $\mathcal{P}(\mathcal{F}^{KD}|\mathcal{I}, y; \theta_T) + \lambda - g$, where $\lambda$ is*

$$\lambda = \mathcal{P}(\mathcal{F}_*^{img}|\mathcal{I}, y; \theta_T) - \mathcal{P}(\mathcal{F}^{KD}|\mathcal{I}, y; \theta_T). \tag{2}$$

*Proof of Lemma 1:*

$$\begin{aligned}
\mathcal{P}(\mathcal{F}^{KD}|\mathcal{P}, y; \theta_S, \mathcal{K}_T) =& \mathcal{P}(\mathcal{F}^{KD}|\mathcal{P}, y; \theta_S, \mathcal{K}_T) - \mathcal{P}(\mathcal{F}_*^{pts}|\mathcal{P}, y; \theta_S, \mathcal{K}_T) \\
&+ \mathcal{P}(\mathcal{F}_*^{pts}|\mathcal{P}, y; \theta_S, \mathcal{K}_T) - \mathcal{P}(\mathcal{F}_*^{img}|\mathcal{I}, y; \theta_T) \\
&+ \mathcal{P}(\mathcal{F}_*^{img}|\mathcal{I}, y; \theta_T) - \mathcal{P}(\mathcal{F}^{KD}|\mathcal{I}, y; \theta_T) + \mathcal{P}(\mathcal{F}^{KD}|\mathcal{I}, y; \theta_T).
\end{aligned} \tag{3}$$

For a successful distillation, the term $\mathcal{P}(\mathcal{F}^{KD}|\mathcal{P}, y; \theta_S, \mathcal{K}_T) - \mathcal{P}(\mathcal{F}_*^{pts}|\mathcal{P}, y; \theta_S, \mathcal{K}_T)$ should be greater than or equal to $0$, *i.e.*, the distilled model outperforms the original point cloud network. By the definition, we have $\mathcal{P}(\mathcal{F}^{KD}|\mathcal{P}, y; \theta_S, \mathcal{K}_T) \geq \mathcal{P}(\mathcal{F}^{KD}|\mathcal{I}, y; \theta_T) + \lambda - g$.

## B  Additional Experiments

### B.1  Comparison with Different Normalization

As we mentioned in the manuscript, the discrepancy between two modalities make the KD problem challenging, which is the initial motivation of our PointCMT. In this section, we analyze different strategies that are used to eliminate the discrepancy. Specifically, we design two normalization:

**Normalize-I**: We assume the features from image and point cloud networks follow two Gaussian distributions. We normalize the mean and standard deviation of the features from image network and make it closer to the features from point cloud network. For every batch of paired image and point cloud data, denote $N$ as batch size, let $\{(\mathcal{F}_i^{pts}, \mathcal{F}_i^{img})\}_{i=1}^N$ be feature pairs from point cloud and image networks. Let mean($\mathcal{F}^{pts}$), std($\mathcal{F}^{pts}$) be the mean and standard deviation of $\{\mathcal{F}_i^{pts}\}_{i=1}^N$, and mean($\mathcal{F}^{img}$) and std($\mathcal{F}^{img}$) be the mean and standard deviation of $\{\mathcal{F}_i^{img}\}_{i=1}^N$. We normalize image features through:

$$\hat{\mathcal{F}}_i^{img} = ((\mathcal{F}_i^{img} - \text{mean}(\mathcal{F}^{img}))/\text{std}(\mathcal{F}^{img})) * \text{std}(\mathcal{F}^{pts}) + \text{mean}(\mathcal{F}^{pts}). \tag{4}$$

**Normalize-II**: We regard features as vectors in a feature space, and normalize their norms into the same scale. Let norm($\mathcal{F}^{pts}$) and norm($\mathcal{F}^{img}$) be the mean of $\{||\mathcal{F}_i^{pts}||_2\}_{i=1}^n$ and $\{||\mathcal{F}_i^{img}||_2\}_{i=1}^n$, where $||\cdot||_2$ denote 2-norm, we normalize image features through:

$$\hat{\mathcal{F}}_i^{img} = (\mathcal{F}_i^{img}/\text{norm}(\mathcal{F}^{img})) * \text{norm}(\mathcal{F}^{pts}). \tag{5}$$

During the experiment, we train PointNet++ with above normalization through KD loss:

$$\mathcal{L}_{KD} = MSE(\mathcal{F}^{pts} - \text{Norm}(\mathcal{F}^{img})), \tag{6}$$

where Norm($\cdot$) denotes the normalization operations mentioned above. As shown in Table 1, exploiting normalization strategies can slightly improve the performance on ModelNet40 and eliminate the negative transfer. Nevertheless, our PointCMT still achieves superior performance upon such naive normalization.

Table 1: Comparison with different normalization on ModelNet40.

| Method | ModelNet40 |
|---|---|
| Baseline | 93.4 |
| Hinton *et al.* | 93.1 (-0.3) |
| w/ Normalize-I | 93.4 (+0.0) |
| w/ Normalize-II | 93.5 (+0.1) |
| PointCMT (ours) | **94.4** (+1.0) |

### B.2  Effect of Pre-trained Image Network

In Table 2, we illustrate the results on ModelNet40 without pre-trained image feature extractor. As shown in the table, when we train image feature extractor from scratch, PointCMT still gains 94.2% overall accuracy, *i.e.*, with only 0.2% performance drop. Moreover, the results on ScanObjectNN are obtained without pre-trained image feature extractor, since the view-images used on this dataset are from perspective projection.

Table 2: Results of using a image network without pre-training on ModelNet40.

| Method | ModelNet40 |
|---|---|
| Baseline | 93.4 |
| PointCMT w/o pre-train | 94.2 (+0.8) |
| PointCMT | **94.4** (+1.0) |

## B.3 Performance of Image Networks

In this section, we demonstrate the performance of image networks pre-trained in the Stage I. As shown in Table 3, through exploiting rendered view-image from CAD models, the image network can achieve very high overall accuracy of 97.0% on ModelNet40, boosting the PointNet++ via PointCMT by a 1% performance gain. Only using projection in the image network can only obtain 93.8% on ModelNet40. Nevertheless, it still increases the performance of PointNet++ by 0.6%. Moreover, we find out that in the cross-modal KD scenario, the performance of the teacher will not always better than the student. Still, PointCMT effectively learn the complementary information from the teacher, and improve the performance of point cloud analysis approaches. For the ScanObjectNN dataset, using additional color information makes image networks overfit on the OBJ_ONLY sub-set, and thus hampers the performance.

Table 3: Results of image networks on ModelNet40 and ScanObjectNN datasets.

| Image Networks | ModelNet40 | OBJ_ONLY | PB_T50_RS |
|---|---|---|---|
| Rendered from CAD | 97.0 | - | - |
| Gains on PointNet++ | (+1.0) | - | - |
| Projection | 93.8 | 89.0 | 80.8 |
| Gains on PointNet++ | (+0.6) | (+4.3) | (+3.9) |
| Projection w/ color | - | 87.5 | - |
| Gains on PointNet++ | - | (+3.2) | - |

## B.4 Training Speed

In this section, we demonstrate the training speed of our PointCMT. As shown in the Table 4, the additional training stage of I (image encoder and image classifier) and II (CMPG) actually introduce little extra cost in the entire training phrase since the small epoch numbers for stage I and few parameters of CMPG. For the speed evaluation of the Stage III, since the image network has been trained, we fixed pre-trained network and generate objects' features offline, which can be directly exploited in the Stage II and III without repeatedly forwarding the image network.

Table 4: The cost of each stage with the form of time for per sample (ms) and total epochs (h).

| Stage I (Image Network) | Stage II (CMPG) | Stage III (PointNet++) |
|---|---|---|
| 27.35ms / 4.36h | 2.3ms / 0.46h | 10.64ms / 36.38h |

## B.5 Part Segmentation

To illustrate the superiority of our PointCMT, we set two experiments for part segmentation on ShapeNetPart dataset, as shown in Table 5. **(a)** PointNet++ with pre-trained encoder (trained from scratch on ModelNet40); **(b)** PointNet++ with pre-trained encoder (trained with PointCMT on ModelNet40). As shown in Table, utilizing pre-trained encoder trained with PointCMT effectively improve the performance, especially for the more challenging metric of Class avg IoU. Here, Inctance avg IoU and Class avg IoU denote the IoU averaged by all instances and each class, respectively.

Table 5: Results on ShapeNetPart with metrics of instance average IoU and class average IoU..

| Method | Inctance avg IoU | Class avg IoU |
|---|---|---|
| Pre-trained PointNet++ w/o PointCMT | 85.3 | 82.0 |
| Pre-trained PointNet++ w/ PointCMT | 85.6 (+0.3) | 82.6 (+0.6) |