# OpenReview forum: "Let Images Give You More: Point Cloud Cross-Modal Training for Shape Analysis"
_NeurIPS.cc/2022/Conference — NeurIPS 2022 Accept_

### Official Review · Reviewer_R3qG · 2022-06-24

**Rating:** 7
**Confidence:** 5
**Soundness:** 3 good
**Presentation:** 3 good
**Contribution:** 3 good

**Summary:**

The authors present a method to enhance point cloud classification models by exploiting training on images. Features extracted from images may be able to capture complementary details to what is typically learned by training on point clouds alone. A distillation procedure is presented so that such complementary features can be integrated in a model that only processes a point cloud at test time.

**Questions:**

- why have chosen the EMD instead of the Chamfer distance as a metric for your loss function?
- it seems you reference the published results for PointMLP. Have you been able to reproduce them? Based on experience and online discussions (https://github.com/ma-xu/pointMLP-pytorch/issues/1), it seems they are quite tricky to reproduce

**Limitations:**

I would like to see an extended discussion about the single-modal setting, in which only point cloud data are used at testing time (although multiple modalities may be available for training) considered in this paper against a fully multi-modal setting.

**Strengths And Weaknesses:**

Overall, the method is sound and generally well explained. A few mistakes in the use of English are present and could be fixed (for example page 2 line 55 "Effectively" --> "Effectiveness").

Strenghts:
- sensible approach to improve feature learning by leveraging complementary features that are more easily extracted from images
- the distillation approach allows to still work in a single-modal setting at test time
- good results showing a benefit from the proposed technique on state of the art architectures
- good ablations to validate each proposed component (CPMG/Feature enhancement)

Weaknesses:
- distillation allows to train the point cloud encoder to extract those features that are easily extracted from images but generally escape classifiers trained on point clouds, despite those features being present in the input point cloud. However, because at test time only the point cloud is used, it is not possible to exploit truly complementary features that an image might carry and that are not present in the point cloud data. This would only be possible in a truly multi-modal setting, also at test time
- the training of CPMG requires paired data with the image and point cloud of the same object. This is somewhat constrasting with the motivation of the work in page 1 line 34 highlighting the potential difficulty in having paired multimodal data

---

> ### Author Response · Authors · 2022-08-01
> **Response to Reviewer R3qG**
>
> We thank the reviewer for the encouraging comments and will further improve this work.
>
> **For Weakness**
>
> * We admire reviewer’s comment that the complementary features from real images cannot be accessed through point clouds in the test time. However, it is not in conflict with our motivation of **improving the performance of point cloud model in the training stage**. **a)** Without additional view or projection images, 3D model sometime cannot learn discriminative features through only using point cloud as input. After conducting regularization through image features,  the features of 3D model will be more diversified and robust.  **b)** Though additional input is not available in the test time, they inherently improve optimization of the 3D model through fully end-to-end training, encoding the prior information in the parameters of the 3D model.
>
> * In L34,  we emphasized  the paired-images are usually unavailable during the **test time**. For instance, we can train point cloud models with PointCMT on existed public dataset with both images and point clouds. However, when we deploy on the real-world cases, such as an indoor robot that only collects point clouds as input, gaining the additional pair-images for the robot seems to be *potentially difficult*.  In this case, PointCMT  give us another choice that enhance point cloud analysis performance in the training phrase. Besides, PointCMT actually acts as an effective training technique that boosts the performance of the point cloud methods, which is also beneficial to multi-modal approaches if models for two modalities can be fused. More importantly, our PointCMT inherently boosts the robustness of single- or multi- sensor systems, especially for multi-sensor damage scenarios.
>
>
> **For Question**
>
> 1. EMD loss is adopted since using CD loss make CMPG converge more slowly.
> 2. We **cannot** reproduce the results of PointMLP on ModelNet40, as depicted in L249-251. However, their results on ScanObjectNN are easier to be reproduced.

---

> > ### Comment · Reviewer_R3qG · 2022-08-09
> > **Thank you for the response**
> >
> > The response addressed my concerns. I will keep my Accept score.

---

### Official Review · Reviewer_vTJ8 · 2022-07-06

**Rating:** 5
**Confidence:** 4
**Soundness:** 3 good
**Presentation:** 3 good
**Contribution:** 2 fair

**Summary:**

This paper proposes a cross-modal training scheme called PointCMT which utilizes both 3D point cloud and synthetically rendered or view projected 2D image to boost point cloud classification performance. The cross-modal training is formulated as a knowledge distillation problem and can be easily combined with existing 3D point cloud based algorithms such as PointNet++ and PointMLP. Experiment shows that PointNet++ with PointCMT can achieve 1.0% and 4.4% accuracy improvements on ModelNet40 and ScanObjectNN benchmark dataset, respectively.

**Questions:**

1. How about the generalization capability of cross-modal training? for example, the performance of training on ModelNet40 and test on ScanObjectNN or vice versa, as similar experiment has been done in [14]

**Limitations:**

There is no potential negative societal impact found

**Strengths And Weaknesses:**

Strength
1. The idea of using view projected 2D image for 3D point cloud classification has been presented in SimpleView[14], but the scheme of cross-modal training using both 3D point cloud and view projected 2D image is novel and interesting.
2. A cross-modal point generator is proposed to solve the cross-modal knowledge distillation problem when the feature distribution of 3D point cloud and 2D image is different and complementary
3. The analysis and proof on the discrepancy between the discriminative image and point cloud features is good.

Weakness:
1. There are some concerns on the experiment:
    a) In table 1, it seems that even equipped with PointCMT, The PointNet++/RS-CNN/DGCNN seems no better than SimpleView [14] which achieves 93.9% overall accuracy with speed of 2208 samples/second, does this mean that cross-modal information fusion (3D point cloud + 2D view image) may not help on the ModelNet40 dataset and only 2D view image is enough?
    b) In table 6, the performance of "with projection" achieves 91.8% accuracy on OBJ_ONLY, which is about +4.3% improvement according to Table 2, but the performance of "with Project with color" only achieves 90.7% accuracy, dropped by 1.1%, does this mean the use of point color information hurts performance of cross-modal training?
2. Since the cross-model training may increase the training time, it would be good to include both training cost and performance gain in the comparison experiment

---

> ### Author Response · Authors · 2022-08-01
> **Response to Reviewer vTJ8**
>
> We thank the reviewer’s beneficial suggestions.
>
> **For Weakness**
>
> 1. **a)** DGCNN and RSCNN with PointCMT cannot beat Simpleview partly because their baselines are too weak (only 92.9%), though PointCMT have already improved some of them by about 1%. Moreover, ModelNet40 dataset is a small synthesis dataset which is easily saturated in performance (reviewer J8fY). This is why we compare our methods on other two more challenging benchmarks on ScanObjectNN dataset, on which PointCMT improves the baseline by over 4%. **b)** The reason why color cannot improve the results is  that OBJ_ONLY dataset only contains 2,902 objects, and image networks are easier to overfit when using the color information. The worse performance of the teacher makes the point cloud model improved little, which has already provided in L335-337.
> 2. We provide concrete training cost and analysis in **Common Reply**.
>
> **For Question**
>
> The model trained with PointCMT also keeps the high generalization ability, and the results are shown in **Table 4**.
>
> **Table 4. Results of generalization test with ModelNet40 to ScanObjectNN (M40 to SONN) and inverse.**
> | Method | M40 to SONN | SONN to M40 |
> |--|--|--|
> | PointNet++ (**official**) |  47.8 | 30.1 |
> | PointNet++ with PointCMT | **49.5** (+ 1.7) | **31.1** (+ 1.0) |

---

> ### Author Response · Authors · 2022-08-09
> **Response to Reviewer vTJ8**
>
> We thank the reviewer’s beneficial suggestions. It seems the deadline of the author-reviewer discussion period. Does our response solve your questions well? Looking forwarding to any feedback.

---

> > ### Comment · Reviewer_vTJ8 · 2022-08-10
> > **The additional experiments have addressed my concerns**
> >
> > Thank for the additional experiment, after reading the author's response and other reviewers' comment, now I raised my score to 5

---

### Official Review · Reviewer_J8fY · 2022-07-06

**Rating:** 5
**Confidence:** 4
**Soundness:** 1 poor
**Presentation:** 3 good
**Contribution:** 1 poor

**Summary:**

This paper shows a 2D-to-3D distillation framework PointCMT that improves 3D classification through cross-modal training. Despite the performance improvement in two classification datasets,  the reviewer has concerns about the effectiveness of PointCMT: (1) the theoretical argument of this approach; (2) some important ablation studies are missed; (3) more serious applications like segmentation should be considered.

**Questions:**

#### Majors:

1. PointCMT might not be theoretically sound. Feature loss (Eqn. 5) will be zero if global features of image and point cloud are equal, since the same CMPT is used for both inputs. Classifier loss (Eqn. 7) will be zero if global features of image and point cloud are equal, since the same classifier (Cls^{pts}) is used. Therefore, the whole regularization term in PointCMT is just to make two global features identical, which means the PointCMT can be replaced by a simple regularization term that makes two features closer to each other.
2. Table 5 needs more clarification. It is not clear which component is ablated. Take row 1 (baseline) and row 2 (Hinton) for example, is it only the logits distillation part be ablated? In other words, are you using eqn. 6 as a replacement of eqn. 7. Or, is the feature distillation ablated such that you are studying the different possibilities of Eqn. 5? Or, both?
3. An ablation study on with/without CMPG should be added if Table 5 is not for it.
4. An ablation study on using Eqn. 6 instead of Eqn. 7 should be added if Table 5 is not for it.
5. An ablation study on values of tradeoff parameters (alpha and beta) in L217 Eqn. should be added. 30 and 0.3 are not common values. How much variance can the parameters cause to the performance? Is that possible that you outperform other distillation techniques simply because the parameters are not good for them in Table 5.
6. Overclaim. There is already a work in 2d-to-3d knowledge distillation. [1]
7. Application is limited to classification only. Classification is the most trivial task in point cloud understanding and the most unimportant task. The classification benchmark datasets in this work are rather small-scale and networks are easily saturated in performance. More serious applications such as segmentation should be considered.

[1]  Yueh-Cheng Liu, etc. Learning from 2D: Contrastive Pixel-to-Point Knowledge Transfer for 3D Pretraining

#### minors:

1. Revise CMPG description. For L181, you should revise to something like: the cross-modal point generator (CMPG) is used to map the global feature representation acquired from images/points into the Euclidean space. Note that CMPG will be used to transform both images and point clouds after pretraining. Since it is the first time you define CMPG, the function of CMPG should be made clear.
2. Better illustration for Fig. 2. Fig.2 (a) dash lines from global feature to feature Enhancement can be illustrated in other colors, \eg blue, since it requires the CMPG module and thus should be highlighted differently for easier understanding. In (b), image features should be changed to global features from images or points.
3. Highlight training cost instead of inference speed. It is weird to highlight the inference speed in L257, since PointCMT only influences training speed and has no effect on inference speed. The inference speed is totally dependent on the point cloud network used in PointCMT. The author should highlight training costs instead.
4. The authors should explicitly highlight that for all experiments, they trained baseline with and without PointCMT using the same optimization techniques, data augmentation, and evaluation techniques. For example, SimpleView found these tricks matter a lot in performance.
5. Any clue why PointCMT improves PointNet++ more than other networks?

**Limitations:**

The authors did not include limitations and potential negative societal impacts. Limitations, e.g. only for classification, can be added.

**Strengths And Weaknesses:**

PointCMT shows cross-modal training that improves performance. Check questions for the weakness.

---

> ### Author Response · Authors · 2022-08-01
> **Response to Reviewer J8fY**
>
> **For Weakness (Majors)**
>
> 1. We respectfully **disagree** with the reviewer’s comment.
> **a)** Since the permutation invariance nature of EMD loss, **the features of two networks are not identical when the EMD loss is equal to zero**. In contrast, we compare multiple traditional knowledge distillation methods that directly make teachers’ and students’ features identical, where our method significantly performs better. Moreover, we also illustrate that simply regularization in the feature space cannot improve the point cloud methods in section B.1 in the supplementary.
> **b)** We formulate the cross-modal learning problem as a knowledge distillation problem, in which the target of most methods is to make the features or outputs of the teacher and student identical. However, due to the discrepancies of data or network architecture, the above problem is generally hard to optimize, and cannot be solved by **a simple regularization term**. This is the reason why previous knowledge distillation methods are proposed and also the motivation of our PointCMT.
> **c)** In spite of the difficulty in optimization, we theoretically provide the lower bound of probability in our formulation and the proof is shown in supplementary material as well.
>
> 2. Thanks for your suggestion, and we will make the illustration of Table 5 clearer. The Table aims to compare the full model of PointCMT (Feature Enhancement and Classifier Enhancement losses) with traditional knowledge distillation techniques using the same **baseline PointNet++**, as depicted in the caption. During the implementation of traditional KD methods, we strictly follow their official paper and apply the full architectures. As shown in table, the traditional KD cannot work well in the cross-modal scenario, i.e., Hinton’s method even makes performance of baseline PointNet++ decrease on ModelNet40 dataset.
> 3. The ablation results of before and after using feature enhancement (FE) though CMPG and classifier enhancement (CE) have already shown in the Table 4 in our manuscript.
> 4. The same of the 3.
> 5. In our experiment, we tune our hyperparameters (alpha and beta) through fix the other to zero. The results are shown in **Table 3**.
>
> **Table 3. Ablated results with different Alpha and Beta on ModelNet40 dataset with PointNet++ baseline.**
> | Alpha (Beta=0)| 1 | 3 | 10 | 30|
> |--|--|--|--|--|
> | OA (%) |  93.4 | 93.5 | 93.5 | **93.8** |
>
> | Beta (Alpha=0) | 0.1 | 0.3 | 1| 3|
> |--|--|--|--|--|
> | OA (%) |  93.7 | **94.0**  | 70.1 | 60.2 |
>
> 6. As discussed in L116-117, though Liu etc [30] was also proposed for cross-modal knowledge transfer, it uses the contrastive learning manner for 3D pre-training that is not relative with our formulation of knowledge distillation.
> 7. We respectfully **disagree** with the reviewer’s comment of **classification is the most trivial task in point cloud understanding and the most unimportant task**. In contrast, it is the most fundamental and important field for point cloud analysis, in which numerous downstream applications are inspired from it. There are several pioneer works only designed for classification [1][2]. However, designed for but not limited to classification as recent works [3][4], the pre-trained model of our PointCMT (e.g., PointNet++ on ModelNet40 and ScanObjectNN) can be used in other downstream applications (e.g., part segmentation on ShapeNetPart and semantic segmentation on ScanNet). Also, we admire reviewer’s perspective that some dataset such as ModelNet40 is easily saturated in performance. Nevertheless, we compare our methods on other two challenging benchmarks on ScanObjectNN dataset, where we improve our baseline PointNet++ from very low accuracy of 79% to 83% with a large improvement of 4%.
>
> [1] Revisiting Point Cloud Shape Classification with a Simple and Effective Baseline, ICML2021
>
> [2] PointCLIP: Point Cloud Understanding by CLIP, CVPR2022
>
> [3] Point-BERT: Pre-training 3D Point Cloud Transformers with Masked Point Modeling, CVPR2022
>
> [4] Masked Autoencoders for Point Cloud Self-supervised Learning, ECCV2022
>
> **For Weakness (Minors)**
>
> We thank the reviewer’s beneficial suggestions. We will make our final version clearer according to the advice. Belows are something we want to emphasize.
> 1. We provide concrete training cost and analysis in **Common Reply**.
> 2. All baseline and baseline with PointCMT **use same training strategies**, which is illustrated in L243-245, eliminating any tricky improvement such as augmentation.
> 3. The improvement of PointNet++ is not more than others (e.g., 1.0 for PointNet++ and 0.9 for RS-CNN) on ModelNet40. For ScanObjectNN dataset, the less improvement of PointMLP is because it contains 10x parameters and already achieves SoTA performance.

---

> > ### Comment · Reviewer_J8fY · 2022-08-06
> > **Thanks the responses from authors. I have following concerns in experiments.**
> >
> > Dear authors, thanks for the reply. It solves my concerns about theoretical arguments.
> > However, you did not convince me about the experiments. I expect experiments in at least ShapeNetPart and fair comparisons with previous methods. Below are the detailed concerns:
> >
> >
> > 1. Table 5 clarification. If I understand correctly, you applied all previous distillation methods on both features and logits part. If so, please add these technical details in the revision and also provide the alphas and betas for the previous methods.
> > 2. Fair comparisons with previous methods. In Rebuttal Table 3, alphas and betas significantly impact the performance. What are the alphas and betas for previous methods? Are they the same as PointCMT.  You did not reply my major concern #5: Is that possible that you outperform other distillation techniques simply because the parameters (alphas and betas) are not good for them in Table 5.  Ablation studies for alphas and betas for previous methods are encouraged. Consider the short rebuttal time, you can work on just one previous method (e.g. Yang etal [57]).
> > 3. Experiments in ShapeNetPart are still missed. Only classification experiments in small datasets are not enough. All experiments in the paper were conducted in very small datasets (ModelNet, ScanObjectNN), where results can be very unstable and random. Even the papers you mentioned (Point-MAE, Point-BERT, etc), all at least showed experiments in part segmentation in ShapeNetPart. I expect the authors show the benefits of the proposed method in this benchmark as well. Your methods can be directly applied there since ShapNetPart contains only objects not scenes, so *why not just show the results*?
> >
> >
> > I **am happy to increase my score to accept** only if I see (i) the **results in ShapeNetPart**, and (ii) **fair comparisons with previous methods** (e.g. play with alphas and betas).

---

> > > ### Author Response · Authors · 2022-08-09
> > > **Results in ShapeNetPart and fair comparisons with previous methods**
> > >
> > > Dear reviewer J8fY
> > >
> > > We appreciate reviewer's encouraging feedbacks and beneficial suggestions. Below are results of two experiments.
> > >
> > > **ShapeNetPart**
> > >
> > > To illustrate the superiority of our PointCMT, we set three experiments as shown in **Table A**. (a) PointNet++ trained from scratch; (b) PointNet++ with pre-trained encoder on ModelNet40; (c) PointNet++ with pre-trained encoder trained by PointCMT on ModelNet40.
> > > As shown in Table, utilizing pre-trained encoder trained by PointCMT effectively improve the performance, especially for the more challenging metric of *Class avg IoU*. Here, *Inctance avg IoU* and *Class avg IoU* denote the IoU averaged by all instances and each class, respectively. These results would be included in the final version.
> > >
> > > **Table A. Results on ShapeNetPart with metrics of instance average IoU and class average IoU.**
> > > | Method | Inctance avg IoU | Class avg IoU |
> > > |--|--|--|
> > > | PointNet++ (official) |  85.1 | 81.9 |
> > > | Pre-trained PointNet++ w/o PointCMT |  85.3 | 82.0|
> > > | Pre-trained PointNet++ w/  PointCMT | **85.6** (+ 0.3) | **82.6** (+ 0.6) |
> > >
> > > **Fair comparison with Yang etal [57]**
> > >
> > > **(1) What are alpha and beta?** As understanding of the reviewer, the alpha and beta are weights for losses of feature and logits alignments.  If the method only has a single alignment, we see the irrelevant hyperparameters as zero, *e.g.,* in Hinton et al., the alpha should be zeros.
> > >
> > > **(2) How to set alpha and beta?** The alpha and beta in our manuscript are chosen by the best hyperparameters in their official papers, *e.g.,* the best hyperparameters of alpha=1, beta=1 in Yang etal [57] on ImageNet. We admire that tuning different alpha and beta would improve the result. We thank reviewer's suggestion and the results in final version would be tuned with different hyperparameters.
> > >
> > > **(3) Results of different alpha and beta.** As the suggestion of the reviewer, we compare results using different alpha and beta in Yang etal [57], which is illustrated in the **Table B**. As shown in the table, tuning different alpha and beta only slightly change the results, especially for the more challenging ScanObjectNN. In our final version, we will tune hyperparameters of Hilton etal and Huang etal with another alpha and beta, and update the results.
> > >
> > > **Table B. Results of Yang etal [57] with different alpha and beta.**
> > > | Method (Alpha, Beta) | OA (ModelNet40) | OA (ScanObjectNN) |
> > > |--|--|--|
> > > | Yang (1, 1) |  93.7 | 81.1 |
> > > | Yang (5, 5) |  93.9 | 81.0 |
> > > | Yang (5, 1) |  93.5 | 80.7 |
> > > | Yang  (1, 5) |  93.8 | 81.1 |
> > > | PointCMT |  **94.4** | **83.3** |

---

> > > > ### Comment · Reviewer_J8fY · 2022-08-09
> > > > **Thanks for the additional experiments**
> > > >
> > > > I appreciate the additional experiments.  I have raised my score to 5.
> > > >
> > > > With regards to ShapeNetPart, please continue working on it, and put the results in revision.
> > > >
> > > > Also, please note that fair comparison with other loss functions is very important, extremely because Rebuttal Table 3 shows the results being **sensitive** to alpha and beta. It is *not fair that you finetune your parameters while leaving other methods using the default parameters* for ImageNet experiments. It is **very possible** that the proposed losses are NOT necessary and the use of Cross-Modal Point Generator is NOT a must. It is also possible that a more fine-grained tuning of previous loss functions (e.g.  Yang. etal) can work well in cross-modal point cloud training.
> > > >
> > > >
> > > > Here are my further concerns: why your methods are sensitive to alpha and beta, while Yang's method does not? Is this the limitation of your methods?

---

> > > > > ### Author Response · Authors · 2022-08-09
> > > > > **Thanks for the feedback**
> > > > >
> > > > > We appreciate reviewer's feedbacks!
> > > > >
> > > > > PointCMT is sensitive to beta (classifier enhancement), but **robust** for alpha (feature enhancement) as previous works. The reason is that, during our classifier enhancement, **the output logits gained by image features are not supervised by the ground truths in the stage III**. Since the classifier is randomly initialized during the stage III, if a very large beta is given, the network will focus on aligning the two output logits but ignoring to regress the ground truths. This makes the network hard to optimize during the initial training phase. Inversely, since the teacher's logits are also supervised by the ground truths in previous works, they have more robust performances with different beta.
> > > > >
> > > > > Though this seems to be a limitation of our method, it can be easily avoided by giving ad-hoc values in a reasonable range, such as [0.1, 0.5].  For instance, we only tune our alpha and beta on ModelNet40 dataset. Nevertheless, when we train the network on the ScanObjectNN without further tuning, it still greatly improves performances on the ScanObjectNN (3.1% and 2.9%), which illustrates that the classifier enhancement works if beta in a reasonable range.
> > > > >
> > > > > If you have any problem, please feel free to let us know.

---

### Official Review · Reviewer_N54s · 2022-07-12

**Rating:** 6
**Confidence:** 3
**Soundness:** 3 good
**Presentation:** 2 fair
**Contribution:** 3 good

**Summary:**

The paper proposes a knowledge distillation strategy to improve point cloud classification. It uses images of point clouds to train an image classification network. The representation from this network can be distilled into any existing point cloud network. The distillation requires multi-stage training including -- training the image classifier, training point generator from images and training the point-cloud network assisted by the image classifier. Experiments show the effectiveness of this scheme.

**Questions:**

Refer to the weakness section for questions. Overall, I am (weakly) positive about the paper. I will update the score based on the rebuttal.

**Strengths And Weaknesses:**

Strengths:

- The paper achieves impressive performance on both the ModelNet40 and ScanObjectNN datasets.

- Gains due to knowledge distillation seem to consistent across various networks (Table 1, Table 2)

Weakness:

- The method archives performance improvements; however, the training pipeline seems to become more complicated with many training stages. This could be a potential limitation. It would be useful if the paper discusses this and potential ways to mitigate it.

- It would be nice if the paper could provide some mean +- std measures. This could be done by running the same experiments multiple times (with random initialization) and reporting the mean and variance. This is particularly important as point-based benchmark methods can have significant variations across runs.

- What is performance of the teacher networks on the two datasets? Also, is the teacher network potentially better because of the additional image to point cloud task? I asking this because as shown in the paper 3D projections could be used and these 3D projections can be created from the point cloud, which is available at test time.

---

> ### Author Response · Authors · 2022-08-01
> **Response to Reviewer N54s**
>
> We thank the reviewer’s careful consideration and beneficial suggestions.
>
> **For Weakness**
>
> * We provide concrete training cost and analysis in **Common Reply**. Moreover, a potential way to reduce the cost of multi-stage training is to jointly train both image and point cloud analysis models in a fully end-to-end manner, which would be investigated in the future work.
> * The measurement of **mean +- std** on ScanObjectNN dataset is provided in **Table 2**, where the current SoTA PointMLP with PointCMT achieves the highest mean with lower std scores compared with their official results. These results would be included in the final version.
> * The results of the teacher network on two datasets have already been shown in **Table 3 (B.3 section) of Supplementary**. As shown in the table, the teacher network does not always perform better than point cloud methods. For instance, on ScanObjetNN dataset, when the teacher only takes projections as inputs, it merely achieves lower accuracy of 80.8% compared with SoTA point cloud methods PointMLP (85.7%). Nevertheless, it still adds an effective regularization upon the point cloud method and brings noticeable improvement (+1.0%). Therefore, PointCMT provides an alternative solution to enhance the point cloud shape analysis when the additional rendered images are not accessible.
>
> **Table 2. The results of PointCMT on ModelNet40 and ScanObjectNN with mean +- std measures.**
> | DataSet | Method | mAcc | OA |
> |--|--|--|--|
> | ModelNet40 | DGCNN w/ Point CMT | 90.5+-0.3 |  93.4+-0.1 |
> | ModelNet40 | RS-CNN w/ Point CMT | 89.9+-0.2 | 93.6+-0.2 |
> | ModelNet40 | PointNet++ w/ Point CMT  | 91.0+-0.2 | 94.2+-0.2 |
> | ScanObjectNN | PointMLP (**official**)  | 83.9+-0.5 | 85.4+-0.3 |
> | ScanObjectNN | PointNet++ w/ Point CMT  | 79.9+-0.3 | 83.1+-0.2 |
> | ScanObjectNN |  PointMLP w/ Point CMT |84.4+-0.4  | 86.4+-0.3|

---

### Author Response · Authors · 2022-08-01
**Common Reply**

We sincerely thank the reviewers’ feedbacks. We will further polish our final version. Below is the response for the common questions.

**Table 1. We demonstrate the training cost of each stage with the form of time for per sample (ms) and total epochs (h).**
| Stage I (Image) | Stage II (CMPG) | Stage III (PointNet++) |
|--|--|--|
| 27.35ms /4.36h |  2.3ms / 0.46h | 5.32ms / 18.19h |


As shown in the **Table1**, the additional training stage of I (image encoder and image classifier) and II (CMPG) actually introduce little extra cost in the entire training phrase since the small epoch numbers for stage I and few parameters of CMPG. Moreover, once the stage I has been trained, we fixed pre-trained image network to generate objects’ features offline, which can be directly exploited in the Stage II and III without repeatedly forwarding the image network.

**Novelty**

We want to re-emphasize our PointCMT is the first to formulate cross-modal learning on point cloud analysis as a knowledge distillation problem, and we theoretically show and prove it lower bound. Through a simple but effective design, PointCMT greatly boosts several baseline models on three datasets **without extra  structural modification, computation burden and data**.

---

### Meta-Review · Area_Chair_NNUn · 2022-08-23

**Recommendation:** Accept
**Confidence:** Certain

**Metareview:**

The paper focuses on the problem of distilling semantic/representation knowledge from 2D images to help further enrich 3D point cloud representation. It received four detailed reviews and a healthy interaction between authors and reviewers ensued. In that back-and-forth, the reviewers clearly stated the weaknesses/issues they saw in the paper, which the authors resolved for the most part through their additional analysis, explanation/clarification, and experiments (e.g. on ShapeNetPart). As such, some reviewers raised their initial review score. Overall, the paper targets an interesting topic in 3D representation learning and it exceeds the bar of contribution and impact expected in NeurIPS papers. The authors are expected to include their additional experiments and discussions in the final version of the paper.

**Award:**

No

---

### Decision · Program_Chairs · 2022-09-14

Accept